# A Study on Rehabilitation Specialists’ Perception of Experience with a Virtual Reality Program

**DOI:** 10.3390/healthcare11060814

**Published:** 2023-03-09

**Authors:** Jung-Ah Lee, Jwa-Gyeom Kim, Hyosun Kweon

**Affiliations:** 1Department of Clinical Rehabilitation Research, Korea National Rehabilitation Research Institute, 58 Samgaksan-ro, Gangbuk-gu, Seoul 01022, Republic of Korea; 2Counselling and Coaching Department, Graduate School, Dongguk University, Seoul 04620, Republic of Korea

**Keywords:** rehabilitation specialists, virtual reality (VR) program, administration experience, Q methodology, subjectivity, rehabilitation therapy

## Abstract

This study aimed to analyse the types and characteristics of rehabilitation specialists’ experience in administering virtual reality (VR) programs using Q methodology as well as provide basic data regarding the introduction of VR programs in rehabilitation therapy. Thirty-three statements were derived based on a literature review and an in-depth interview with rehabilitation specialists. Q-sorting was performed by a Q-sample of 22 rehabilitation specialists with experience administering VR programs. Data were analysed using the QUANL Program. Rehabilitation specialists’ experience with administering VR programs were classified into three types: type 1 was labelled ‘the need to develop VR programs customised for disability level and type’, type 2 was labelled ‘emphasis on experts’ role of paying attention continuously and their experience’, and type 3 was labelled ‘the need to develop safety equipment by disability type’. The study’s findings demonstrate that there are a variety of rehabilitation specialists’ perceptions on their experience administering VR programs. This study is of significance because it suggests ways to improve VR programs for disabled people, with the consultation of rehabilitation specialists. In addition, rehabilitation specialists’ perceptions on VR programs have not been explored before.

## 1. Introduction

An important technology within the fourth industrial revolution is virtual reality (VR). This refers to a human-computer interface in which a specific environment and situation is created to enable users to experience it as if they were interacting with a real environment [1]. Recently, the application of VR technology has been expanded by various media, and it is predicted that the technology will rapidly advance owing to the development of information technology and the wide spread of smartphone usage.

VR has been utilised in fields such as healthcare, education, rehabilitation, industries, and games [2,3,4,5,6,7]. In particular, studies have been conducted to develop various programs for rehabilitation of the elderly and the disabled to investigate the effects [8,9,10,11,12,13,14]. Hence, based on the strengths of VR technology, its applicability as a rehabilitation and therapeutic tool is vast [15]. Conventional rehabilitation therapy is performed in the presence of therapists within a single environment, whereas VR-based rehabilitation therapy can help build a foundation of telerehabilitation. This is on account of VR using technology, which enables patients to continue to receive rehabilitation therapy in a constant condition through a variety of virtual environments rather than a single environment [16,17]. VR-based rehabilitation therapy can provide a wide range of patients with an opportunity for treatment and can be more effective. An additional strength of the therapy is that VR environments can be modified to meet the needs of disabled individuals [18].

The aim for the disabled person is social security practice, with the objectives of ‘independent living’ and ‘normalisation’. Preparation of the disabled person for independence, such as rehabilitation training, is an essential component for living in a community. Through rehabilitation training, the disabled person can have jobs and form social networks via interpersonal interaction at work, which then leads to opportunities for social participation.

To live independently, the disabled person can receive a variety of rehabilitation training appropriate to their circumstances in rehabilitation hospitals and centres, as well as an education. Through rehabilitation training to cope with society, the disabled person can prepare themselves for a social life and communicate with others in daily life [19]. Despite interest in helping disabled individuals and the introduction of a variety of systems, there are environmental barriers in Korean society [20]. Accordingly, rehabilitation activities for people who are disabled are crucial even after social return, and to remain healthy, an effort to improve health management competency by participating in various programs is essential.

Rehabilitation is not a service provided only for the individuals with physical disabilities or organ failure, but a critical element in effective health management for those with acute or chronic pain and functional disorders. As such, rehabilitation should be available to a wider population. Hence, in a broad sense, rehabilitation is a health management strategy and is more effective if it is accompanied with a personalised management strategy and performed with essential assistive devices [20,21,22,23,24].

It is rehabilitation specialists whose roles consist of championing programs appropriate for disabled individuals and supporting them to improve quality of life, functional rehabilitation, and social participation, aside from physical rehabilitation [22,25]. Rehabilitation specialists include physical therapists and occupational therapists [26]. They take on a variety of roles in the frontline of rehabilitation therapy for the disabled and directly experience the process where new rehabilitation therapy programs are developed and applied and see the programs’ effect in helping people with disabilities [27,28]. Therefore, if VR programs are developed based on rehabilitation specialists’ perception of experience with VR program administration, prior to full application in the field, it may be possible to intervene more effectively in challenges experienced by the disabled.

Although VR programs are presently administered to disabled people and shown to be effective, research has not been conducted to investigate rehabilitation specialists’ (who have an important role in the field of rehabilitation therapy) perception of experience with VR program administration. As a core factor explaining the perception (such as personal attitude, feeling, and emotion) of experience with VR program administration, the structure of subjectivity and understanding in rehabilitation specialists is necessary for creating appropriate programs for disabled individuals.

Q methodology is a useful method to uncover individuals’ characteristics and subjectivity and identify difference among types of individuals. This methodology has been evaluated and found to be useful in classifying, analysing, and gaining an understanding of the structure of subjectivity, with a purpose of hypothesis generation rather than hypothesis test and generalisation [29,30]. Q methodology makes it possible to explore the subjectivity of rehabilitation specialists’ experience with VR program administration.

Accordingly, the current study aimed to typify the perception of rehabilitation specialists working at the Korea National Rehabilitation Center (KNRC), regarding their experience of administrating VR programs and to identify the characteristics of each type by using Q methodology. This was to suggest strategies to advance rehabilitation therapy by considering the specialists’ opinions on the utilisation of VR programs and contribute to the improvement of VR program quality.

Specific research questions were as follows:(1)What are the types of subjectivity in rehabilitation specialists’ experience with VR program administration?(2)What are the characteristics of each type of subjectivity in rehabilitation specialists’ experience with VR program administration?

## 2. Methods

### 2.1. Study Design

The purpose of the current study was to identify the types and characteristics of rehabilitation specialists’ subjective perception of VR programs by using Q methodology. Q methodology is a method to uncover and explain individuals’ subjective tendencies and values. It is a simultaneously qualitative and quantitative method to analyse individuals’ operant or self-referent subjectivity [31] and can be used in elucidating individuals’ subjective viewpoints on a specific subject matter [32]. Q methodology has significance in that it can derive diverse subjective judgments, attitudes, meanings, and values. It refers to a methodology of studies designed to examine subjectivity and pursue an in-depth understanding of the subject-matter [30,33].

Based on the features of Q methodology, this study’s researchers determined that it was the most appropriate method for the current study, the aims of which were to explore a variety of subjective perceptions of experience with VR program administration and identify the types of perception by interpreting diverse intrinsic meanings. Particularly, Q methodology was appropriate in this study because an objective of the methodology is to examine individuals’ subjectivity in viewing a specific subject-matter by studying their implicit beliefs, values, and attitudes [34].

Study participants were rehabilitation specialists with experience administering augmented VR programs (rehabilitation sports simulators such as rowing, skiing, wingsuit flying, horse riding, and cycling) as well as immersive VR programs (interacting with floors and walls) 10-times or more. Both programs required head mount displays (HMD) for program participation.

In the study, according to the Q methodology procedure, a Q-population was constructed and from this population came a Q-sample and then a P-sample to sort the Q-sample [29]. The study procedure, including data analysis following the Q methodology procedure, is as shown in Figure 1.

### 2.2. Selection of Q-Sample and P-Sample

#### 2.2.1. Q-Population Construction and Q-Sample Selection

A Q-population refers to Q statements related to the study topic. Q statements are collected primarily through in-depth interviews with experts on the topic, ordinary people, as well as a literature review [35]. To construct a Q-population in this study, a literature review was performed, and in-depth interviews were conducted with 5 rehabilitation specialists who had experience administering augmented and immersive VR programs 10 times or more. To construct statements, first, the researchers derived a total of 53 statements regarding rehabilitation specialists’ experience with VR program administration, from news articles and previous studies [10,11,12,36,37,38]. Next, 5 rehabilitation specialists were interviewed and as a result, 21 statements were added, therefore there was a total of 74 statements.

To select the Q-sample, 74 statements in the Q-population were iteratively read. Redundant statements were revised and integrated. To ensure the validity of the statements, a Q-methodology expert, a rehabilitation specialist experienced in augmented and immersive VR program administration, and a hands-on worker in rehabilitation with experience in VR program administration were consulted. Of the statements, the 33 which were determined to better reflect the experience were selected for the Q-sample. The 33 statements are shown in Table 1.

#### 2.2.2. P-Sample Selection

P-sample refers to the sample of a small number of participants who know the study topic and perform a Q-sort task [39]. In general, the R methodology selects a large number of participants as samples, but the Q methodology selects a relatively small number of participants as samples [31]. When the size of the P sample increases, there is a problem that many people are classified under one factor, and the characteristics of the factor become unclear. Therefore, the number of P samples in the Q methodology can be compared between factors, and it is sufficient to make factors [29,40]. In addition, the P sample is appropriate for around 20 to 30 people, and these small groups are considered to represent a specific group [29]. Therefore, a total of 22 participants was selected as rehabilitation experts at the National Rehabilitation Center for the purpose of sampling for this study. The P sample consisted of rehabilitation experts with experience in running a virtual reality program, and 3 occupational therapists, 10 physical therapists, 5 rehabilitation sports instructors, 2 social workers, and 2 clinical rehabilitation researchers participated as P samples of this study. The average experience of rehabilitation experts was 11 years and information regarding study participants is shown in Table 2.

#### 2.2.3. Q-Sorting

Q-sorting refers to a task the P-sample performs to sort Q-statements using a Q-sort distribution grid [40]. In this study, participants (P-sample) were instructed to read the statements written on cards (Q-sample) and sort all cards using the Q-sample distribution grid (the forced distribution method). These methods are traditionally preferred methods in the study of Q methodology [41]. The Q sample classification diagram is shown in Figure 2.

Specifically, participants (P-sample) were instructed to sort the 33 statements (Q-sample) to the most agreed, the most disagreed, or neutral, based on their subjective judgements and then, fill in the blanks in the Q-sample distribution grid with the statements from the farthest left (−4) or the farthest right (+4) toward the centre (0).

#### 2.2.4. Data Analysis

Q-sort data obtained from P-sample were converted into scores using a 9-point scale. The coding was performed by assigning 1 point to most disagreed items (−4), 9 points to most agreed items (+4), and 5 points to neutral items (0). The distribution chart of Q classification depends on the number of statements, and if the number of statements is 40 or less, a 9-point scale is generally used [31]. Coded data were submitted to principal component analysis using PC-QUANL. Factors were extracted with a criterion of the eigenvalue 1.0 or higher, and z-scores were used to select items fitting to each factor. In addition, factors and correlations were analysed with Varimax rotation. As a result, three types with unique characteristics were classified.

## 3. Results

### 3.1. Analysis Results

The results of principal component analysis using the PC-QUANL Program were as follows. Rehabilitation specialists’ perceptions of experience in VR program administration were sorted into three types as shown in Table 3. The eigenvalue was 8.2752 in type 1, 1.8016 in type 2, and 1.4085 in type 3, and cumulative variance was 0.5221.

Regarding correlation between types, that is, a measure of the degree of similarity of the types, the correlation coefficient was 0.566 between types 1 and 2, 0.627 between types 1 and 3, and 0.470 between types 2 and 3 (Table 4).

Table 5 shows factor weight by type. The participant with the factor weight in each type was P15 (factor weight = 1.5406) in type 1, P1 (factor weight = 1.7204) in type 2, and P4 (factor weight = 1.6786) in type 3.

### 3.2. Characteristics of Each Type of Rehabilitation Specialists’ Perception of Experience in VR Program

#### 3.2.1. Type 1: The Need to Develop VR Programs Customised for Disability Level and Type

Twelve participants were classified to type 1. They had 13 years of experience, on average. Their perception of experience with VR program administration was as shown in Table 6.

Type 1 agreed the most to the following statements: ‘I think that programs for disabled individuals should be appropriate to their levels’ (z = 1.77), ‘I think that VR programs should be designed to be specific to each disability type’ (z = 1.66), ‘I think that VR programs reflecting the needs of disabled individuals should be developed’ (z = 1.61), and ‘I think that VR programs with a variety of sports contents specifically targeting disabled individuals should be developed’ (z = 1.22).

P15 who showed the highest factor weight in type 1, 1.5406, was a physical therapist with 15 years of experience. P15 stated that ‘because the level of disability is different in different individuals, I think that like all other rehabilitation programs, diverse programs reflecting a wide range of disability level should be provided to increase program effect’. P11 (factor weight = 1.2831), an occupational therapist with 11 years of experience, stated that ‘programs designed by considering disability type should be provided to better motivate participants and increase treatment effect. Thus, I think that we need customised programs with the characteristics of each disability type to be taken into account’.

Rehabilitation therapists in type 1 perceived that VR programs should reflect disability type, need, and level of disabled individuals and so, this type was labelled ‘the need to develop VR programs customised for disability level and type’.

#### 3.2.2. Type 2: Emphasis on Experts’ Role in Paying Attention Continuously and Their Experience

Five participants were classified to type 2. They had 9 years of experience, on average. Their perception of experience with VR program administration is as shown in Table 7.

Type 2 agreed the most to the following statements: ‘I think that sensitivity of experts is essential because patients’ attitude toward program participation varies depending on their condition’ (z = 1.60), ‘I think that therapists with a lot of experience with disabled individuals should provide VR programs’ (z = 1.44), and ‘I think that to motivate disabled individuals to participate in VR programs, instructors should continue to pay attention and communicate with them’ (z = 1.36).

P1 who showed the highest factor weight in type 2, 1.7204, was an occupational therapist with 8 years of experience. P1 stated that ‘I think the therapist’s experience is most critical in program application’. P12 (factor weight = 1.1602), a rehabilitation sports instructor with 3 years of experience, said that ‘I think that to achieve the goal of disabled individuals’ participation in sports and exercise, VR programs participation should be consistent and continuous. Hence, I think it the most essential for rehabilitation specialists is to continue to pay attention to induce participants’ interest and select programs designed to achieve the goal’. P12 added that ‘exercise performance and participation enthusiasm vary day by day, depending on the patients’ condition and so, therapists should be sensitive to patient condition and induce active participation’.

As shown above, rehabilitation specialists in type 2 were characterised by their emphasis on the expertise and roles of therapists in providing VR programs to disabled individuals and so, type 2 was labelled ‘emphasis on experts’ role in paying attention continuously and their experience’.

#### 3.2.3. Type 3: The Need to Develop Safety Equipment by Disability Type

Five participants were classified to type 3. They had 9 years of experience, on average. Their perception of experience in VR program administration is shown in Table 8.

Type 3 rehabilitation specialists agreed the most to the following statements: ‘I think that VR programs should be designed to be specific to each disability type’ (z = 2.18), ‘I think that equipment easy for disabled individuals to wear should be developed’ (z = 1.38), ‘I think that it is mandatory to provide safety equipment when disabled individuals participate in VR programs’ (z = 1.31), and ‘I think that it will be more effective if equipment specialised for each disability type is provided’ (z = 1.28). P20 who showed the highest factor weight in type 3, 1.0349, and was a rehabilitation sports instructor who stated that ‘safety problem occurs during VR programs for various reasons. I think that there should be a safety system to predict all such situations’ and that ‘after having tried VR program applications, regarding program applicability by disability type, I think that users and instructors both can be satisfied if equipment is developed for each and every disability type’. P5 (factor weight = 0.6823), a rehabilitation sports instructor with 8 years of experience, stated that ‘I predict that the participation rate will be low and disabled individuals will not even try, because I think wearing the equipment causes a hassle or is dangerous’.

As shown above, rehabilitation specialists in type 3 had the opinion that the effect of VR programs would be greater if safety equipment and the equipment essential for each disability type were developed and utilised during the programs. In this regard, type 3 was labelled ‘the need to develop safety equipment by disability type’.

## 4. Discussion and Conclusions

### 4.1. Discussion

This study was conducted to explore rehabilitation specialists’ subjective perception of experience in VR program administration using Q methodology and identified the following three types: ‘the need to develop disability-customised programs’, ‘emphasis on experts’ role’, and ‘the need to develop safety equipment’. Based on the findings, major characteristics of rehabilitation specialists’ perception of experience with VR program administration will be discussed pertaining to each research question.

First, type 1, labelled ‘the need to develop VR programs customised for disability level and type’, was characterised by the opinions that programs should be provided appropriately for disability type and level and that VR programs should be developed by reflecting the needs of people with disabilities. In addition, rehabilitation specialists in type 1 thought that to increase the program effect, VR programs should reflect participants’ functional level because disability level differs in different individuals and that participants will be motivated to participate in programs in which the type of disability is taken into consideration. In other words, type 1 rehabilitation specialists perceived the need for VR programs to reflect the type and level of disability and the needs of disabled individuals. This finding is in line with a previous study’s findings that the development of programs customised for disability level, the characteristics of disabled individuals, and meeting the demand of the time should be included in VR [42]. Additionally, VR-based programs with the characteristics of the disabled considered had positive impact in cognitive, psychological, and physical areas [43].

Type 1 rehabilitation specialists said that even though programs should be provided based on the consideration of various factors, including participants’ functional level and disability type, current programs administered to the disabled were not specialised for them and so, it was not possible to provide programs customised for disabled individuals. Accordingly, to leverage the strengths of VR programs while decreasing risk and ultimately, to contribute to the development of rehabilitation and social return of people with disabilities via VR programs, the needs of disabled individuals should be identified via usability assessment (e.g., using the System Usability Scale or performing focus group interview (FGI)) and needs-based VR programs developed. This is consistent with previous studies showing high levels of satisfaction as a solution for rehabilitation and treatment of people with brain lesion and securing positive and effective data for patients through continuous clinical trials and usability evaluation [44]. In addition, it is consistent with previous studies that showed virtual reality technology in specialised rehabilitation areas for people with stroke and Parkinson’s disease and helping their daily life [45,46,47,48,49].

Second, type 2, labelled ‘emphasis on experts’ role in paying attention continuously and their experience’, was of opinion that rehabilitation specialists should be sensitive because disabled individuals’ attitude toward program participation varies according to their condition and that VR programs should be administered by highly experienced therapists. In addition, type 2 thought that rehabilitation specialists’ experience was important in VR program application and participation in VR programs required consistency and continuity, accordingly the specialists should continue to pay attention to their patients. In other words, rehabilitation specialists in type 2 seemed to think that the expertise of therapists was essential for administering VR programs to disabled individuals and their role was critical. This finding is consistent with previous study findings that the levels of rehabilitation specialists and service were directly linked with each other [50]. In addition, personal characteristics of people with disabilities using rehabilitation centres somewhat strongly influenced rehabilitation specialists’ competency, as well as social and regulatory environments [51]. Furthermore, it is consistent with previous studies in which the relationship between people with disabilities and experts providing learning to people with disabilities influenced motivation and motivation influenced learning [52].

Type 2 rehabilitation specialists stated that because the therapists’ experience would impact disabled individuals participating in VR programs, highly experienced and adequately educated therapists should administer the programs. Hence, to improve program quality, rehabilitation specialists should be educated to increase expertise in VR program administration. Furthermore, requirements to nationally register and certify practitioners in order to be qualified and administer programs to people with disabilities should be considered. This is consistent with the findings that the role of competent supervisors and experts plays a decisive role in the rehabilitation of the people with disabilities [53] and shows that services are more effective when provided through the cooperation of experts in each field for them [54].

Third, type 3, labelled ‘the need to develop safety equipment by disability type’, had the opinion that it would be necessary to develop equipment for VR programs for disabled individuals to wear and to introduce equipment customised by disability type. Additionally, type 3 rehabilitation specialists thought that a safety system should be in place to predict safety problems in VR program applications and that users and therapists both would be satisfied if safety equipment was developed by disability type. That is, they perceived that the effect of VR programs would be greater if safety problems that might occur during VR programs could be prevented, and safety equipment customised for each disability type be available. This finding is in line with previous studies findings that users may refuse to wear VR devices or have difficulty wearing them [21]. Furthermore, it would be important to provide safety guidelines and strictly manage issues pertaining to burns and vision problems, due to light and heat emission from the VR devices worn on the head [55]. Additionally, the finding is consistent with previous findings that found users can experience nausea and dizziness called cyber or VR sickness [56], therefore the use of VR programs decreased due to discomfort and repulsion of wearing devices because of physical restriction [38].

Rehabilitation specialists in type 3 stated that currently, for various reasons, safety problems occurred during the administration of VR programs and so, both disabled individuals and rehabilitation specialists would be satisfied if safety systems were to be introduced for risk prevention and equipment specialised for each disability type was available. It will be possible to create safer VR programs and specialised for a wide range of disability type, if rehabilitation specialists administering VR programs discuss with VR hardware experts and software experts when developing VR programs.

As discussed above, the existence of rehabilitation specialists’ various perceptions demonstrates the importance for the specialists to self-manage, in order to effectively administer VR programs to people with disabilities. Further, it suggests that it is important to examine the extent of using a checklist and that continuous and professional management at organisation level is necessary [57,58,59]. The current study findings suggest that a satisfaction survey and interview be performed on a regular basis with disabled individuals and rehabilitation specialists administering VR programs to explore the needs of people with disabilities regarding VR programs. This will reflect their needs in the programs and practical approaches for people with disabilities can be identified in consultation with experts in VR technical development [60].

In this study, types, and characteristics of rehabilitation specialists’ subjective perception of experience with VR program administration were analysed by using Q methodology to explore the experience of administering VR programs and provide basic data needed for quality improvement and follow-up research regarding VR programs in rehabilitation therapy.

### 4.2. Limitations

Below, the limitations of the study and suggestions for follow-up studies are discussed.

First, the demographic background of the rehabilitation specialists was not sufficiently discussed. Rehabilitation specialists’ subjectivity with respect to the experience with VR program administration may vary according to the area of expertise, level of experience, and the number of times of having administered VR programs. Second, subjective perception types and characteristics of rehabilitation specialists’ experience in VR program administration were identified by using Q methodology and so, the study findings are limited in generalisability, because unlike R methodology, Q methodology assesses subjectivity based on a small sample of participants randomly selected by the researcher [61]. A third limitation is regarding the use of Q-sorting with forced distribution method—the meaning may differ according to participant, even if different participants place statements in the same positions in the distribution grid.

Based on the current study, it is suggested that follow-up research should be performed by varying rehabilitation specialists’ background, such as the number of times of having administered and provided VR programs and the level of experience. Additionally, it is believed that a dissemination of each of the perception types derived in this study will help rehabilitation specialists provide disabled individuals with high quality VR programs and diverse VR programs can be developed to increase accessibility.

## 5. Conclusions

This study is significant because the structure of rehabilitation specialists’ subjective perception of experience in program administration was classified and examined with focus on VR programs. The study’s findings suggested ways to improve VR programs and laid a foundation for practical directions in program administration. The study’s findings demonstrate that there are various types of perceptions amongst rehabilitation specialists regarding VR program administration and that diverse needs exist. The researchers made suggestions for responding to each perception type and hope that the findings will be useful in providing basic data to identify disabled individuals’ needs and improve their access to VR programs to advance their rehabilitation therapy.

## Figures and Tables

**Figure 1 healthcare-11-00814-f001:**
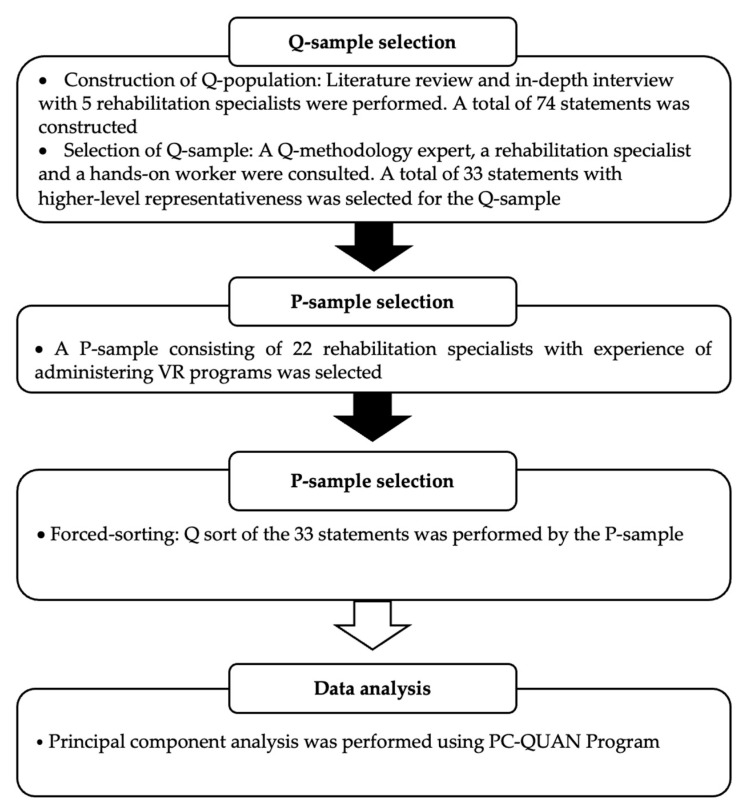
Study procedure.

**Figure 2 healthcare-11-00814-f002:**
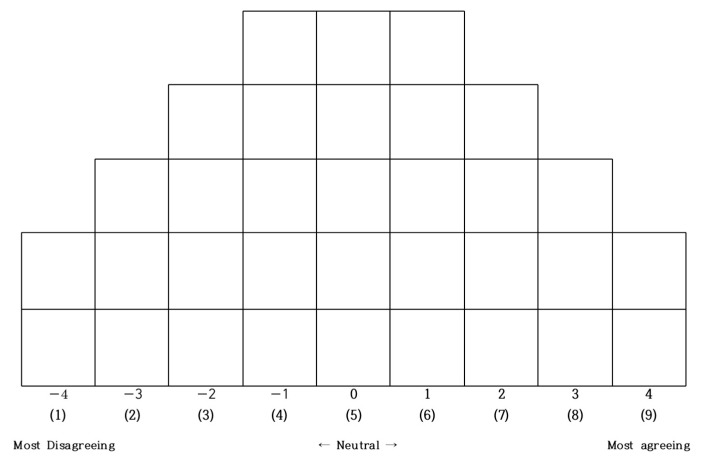
A diagram of Q-sample sorting.

**Table 1 healthcare-11-00814-t001:** Q statements.

ID	Statement
Q1	I think that VR programs should be designed to be specific to each disability type.
Q2	I think that therapists with a lot of experience with disabled individuals should provide VR programs.
Q3	I think that it is complicated to wear the equipment for VR programs.
Q4	I think that VR programs will be effective as an assistive device in rehabilitation and return to society.
Q5	I think that VR programs with a variety of sports contents, specifically targeting disabled individuals should be developed.
Q6	I think that the institutions providing VR programs should share information regarding the programs.
Q7	I think that VR programs reflecting the needs of disabled individuals should be developed.
Q8	I think that to motivate disabled individuals to participate in VR programs, instructors should continue to pay attention and communicate with them.
Q9	I think that differentiated VR programs should be provided each day of a week.
Q10	I think that a disadvantage of VR programs is that they are provided to a relatively small number of individuals compared to the level of manpower and budget needed to create the program.
Q11	I think that equipment easy for disabled individuals to wear should be developed.
Q12	I do not think that VR programs with simple repetitive contents are useful for disabled individuals.
Q13	I think that there should be a separate space dedicated for VR programs.
Q14	I think that VR programs with clear objectives should be created.
Q15	I think that it is unsafe to provide disabled individuals with VR game programs.
Q16	I think that it should be possible for multiple individuals to simultaneously participate in VR programs offline.
Q17	I think that sensitivity of experts is essential because patients’ attitude toward program participation varies depending on their condition.
Q18	I think that we need VR programs incorporating the concepts of reinforcement and reward.
Q19	I think that the qualification and continuing education of therapists responsible for VR programs should be nationally managed.
Q20	I think that VR programs will improve patients’ ability to perform activities of daily living.
Q21	I think that VR programs are limited in that they cannot be offered remotely.
Q22	I have experienced difficulty with VR programs, due to problems manipulating the equipment or equipment malfunctioning.
Q23	I think that programs for disabled individuals should be appropriate to their levels.
Q24	I think that it is safer to provide AR to disabled individuals, because of the risks of VR.
Q25	I think that a program should be developed to accumulate data regarding improved functioning via the use of VR programs.
Q26	When administering VR programs, I have felt confused regarding a therapist’s role.
Q27	I think that one-time participation in a VR program is ineffective.
Q28	I think that it is mandatory to provide safety equipment when disabled individuals participate in VR programs.
Q29	I think that it will be more effective if equipment specialised for each disability type is provided.
Q30	I think that VR programs are useful because participants can meet and interact with others online to increase sociality.
Q31	I think that VR programs should be renewed on a regular basis.
Q32	I think that research on VR programs has not been sufficiently conducted.
Q33	I think that VR programs in rehabilitation centres have not been sufficiently promoted.

**Table 2 healthcare-11-00814-t002:** P sample.

P Sample	Occupation	Experience (Years)
P1	Occupational therapist	8
P2	Clinical rehabilitation researcher	4
P3	Physical therapist	2
P4	Physical therapist	15
P5	Rehabilitation sports instructor	8
P6	Rehabilitation sports instructor	10
P7	Rehabilitation sports instructor	17
P8	Social worker	27
P9	Social worker	5
P10	Physical therapist	6
P11	Occupational therapist	11
P12	Rehabilitation sports instructor	3
P13	Clinical rehabilitation researcher	6
P14	Physical therapist	20
P15	Physical therapist	15
P16	Physical therapist	14
P17	Physical therapist	20
P18	Physical therapist	13
P19	Occupational therapist	17
P20	Rehabilitation sports instructor	4
P21	Physical therapist	20
P22	Physical therapist	5

**Table 3 healthcare-11-00814-t003:** Eigenvalues and explained variances in the 3-type classification.

Contents/Types	I	II	III
Chosen Eigenvalues	8.2752	1.8016	1.4085
Total Variance	0.3761	0.0819	0.0640
Cumulative	0.3761	0.4580	0.5221
Solution Variance	0.7205	0.1569	0.1226
Cumulative	0.7205	0.8774	1.0000

**Table 4 healthcare-11-00814-t004:** Correlations.

Type	I	II	III
I	1.000		
II	0.566	1.000	
III	0.627	0.470	1.000

**Table 5 healthcare-11-00814-t005:** Participants and factor weight by type.

Type 1 (N = 12)			
P Sample	Factor Weight	Occupation	Experience (Years)
P2	1.3181	Clinical rehabilitation researcher	4
P6	0.8555	Rehabilitation sports instructor	10
P7	0.8556	Rehabilitation sports instructor	17
P9	0.6251	Social worker	5
P11	1.2831	Occupational therapist	11
P14	1.3517	Physical therapist	20
P15	1.5406	Physical therapist	15
P16	0.5196	Physical therapist	14
P17	0.8702	Occupational therapist	20
P19	1.2303	Occupational therapist	17
P21	0.6454	Physical therapist	20
P22	0.4804	Physical therapist	5
**Type 2 (N = 5)**			
**P-Sample**	**Factor Weight**	**Occupation**	**Experience (Years)**
P1	1.7204	Occupational therapist	8
P3	1.0896	Physical therapist	2
P8	0.8180	Social worker	27
P12	1.1602	Rehabilitation sports instructor	3
P13	0.9579	Clinical rehabilitation researcher	6
**Type 3 (N = 5)**			
**P-Sample**	**Factor Weight**	**Occupation**	**Experience (Years)**
P4	1.6786	Physical therapist	15
P5	0.6823	Rehabilitation sports instructor	8
P10	0.9642	Physical therapist	6
P18	1.1210	Physical therapist	13
P20	1.0349	Rehabilitation sports instructor	4

**Table 6 healthcare-11-00814-t006:** Type 1 statements and standardised scores (±1.00 and above).

ID	Statement	Standardised Score
23	I think that programs for disabled individuals should be appropriate to their levels.	1.77
1	I think that VR programs should be designed to be specific to each disability type.	1.66
7	I think that VR programs reflecting the needs of disabled individuals should be developed.	1.61
14	I think that VR programs with clear objectives should be created.	1.45
5	I think that VR programs with a variety of sports contents specifically targeting disabled individuals should be developed.	1.22
29	I think that it will be more effective if equipment specialised for each disability type is provided.	1.17
24	I think that it is safer to provided AR to disabled individuals, because of the risks of VR.	−1.08
26	When administering VR programs, I have felt confused regarding a therapist’s role.	−1.28
21	I think that VR programs are limited in that they cannot be offered remotely.	−1.30
15	I think that it is unsafe to provide disabled individuals with VR game programs.	−1.38
12	I do not think that VR programs with simple repetitive contents are useful for disabled individuals.	−1.52
9	I think that differentiated VR programs should be provided each day of a week.	−1.53

**Table 7 healthcare-11-00814-t007:** Type 2 statements and standardised scores (±1.00 and above).

ID	Statement	Standardised Score
17	I think that sensitivity of experts is essential because patients’ attitude toward program participation varies depending on their condition.	1.60
28	I think that it is mandatory to provide safety equipment when disabled individuals participate in VR programs.	1.52
2	I think that therapists with a lot of experience with disabled individuals should provide VR programs.	1.44
8	I think that to motivate disabled individuals to participate in VR programs, instructors should continue to pay attention and communicate with them.	1.36
23	I think that programs for disabled individuals should be appropriate to their levels.	1.28
7	I think that VR programs reflecting the needs of disabled individuals should be developed.	1.16
11	I think that equipment easy for disabled individuals to wear should be developed.	1.14
3	I think that it is complicated to wear the equipment for VR programs.	1.04
33	I think that VR programs in rehabilitation centre have not been sufficiently promoted.	−1.00
9	I think that differentiated VR programs should be provided each day of a week.	−1.16
26	When administering VR programs, I have felt confused regarding a therapist’s role.	−1.27
15	I think that it is unsafe to provide disabled individuals with VR game programs.	−1.60
19	I think that the qualification and continuing education of therapists responsible for VR programs should be nationally managed.	−1.88
24	I think that it is safer to provide AR to disabled individuals due to the risks of VR.	−1.94

**Table 8 healthcare-11-00814-t008:** Type 3 statements and standardised scores (±1.00 and above).

ID	Statement	Standardised Score
1	I think that VR programs should be designed to be specific to each disability type.	2.18
11	I think that equipment easy for disabled individuals to wear should be developed.	1.38
7	I think that VR programs reflecting the needs of disabled individuals should be developed.	1.33
28	I think that it is mandatory to provide safety equipment when disabled individuals participate in VR programs.	1.31
29	I think that it will be more effective if equipment specialised for each disability type is provided.	1.28
23	I think that programs for disabled individuals should be appropriate to their levels.	1.12
19	I think that the qualification and continuing education of therapists responsible for VR programs should be nationally managed.	−1.22
2	I think that therapists with a lot of experience with disabled individuals should provide VR programs.	−1.28
18	I think that we need VR programs incorporating the concepts of reinforcement and reward.	−1.34
12	I do not think that VR programs with simple repetitive contents are useful to disabled individuals.	−1.85
15	I think that it is unsafe to provide disabled individuals with VR game programs.	−2.22

## Data Availability

The data presented in this study are available on request from the corresponding authors.

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
