# Peer review of "A Study on Rehabilitation Specialists’ Perception of Experience with a Virtual Reality Program"

_healthcare, 2023, doi:10.3390/healthcare11060814_

Round 1

Reviewer 1 Report

Good article and novel topic, but I think some small modifications would be necessary.

Within the results I would make several changes:

- I can't locate table 5

- Table 3 is more of a figure or illustration than a table of results as such.

- In the tables I miss the deviations of the weight averages, in addition the correlations should be included in the tables, so that they are more visual.

Within the discussion, I would highlight the Strengths of the study, since the limitations are too extensive, defend why this study is interesting for the scientific community.

Within the Conclusion, it would begin by responding to the stated objective.

Author Response

1) I can't locate table 5

Authors response: Thank you for this comment. We double checked and re-ordered the tables. Please refer to p. 5 to 11.

2) Table 3 is more of a figure or illustration than a table of results as such.

Authors response: We have revised accordingly. Please refer to p.7.

3) In the tables I miss the deviations of the weight averages, in addition the correlations should be included in the tables, so that they are more visual.

Authors response: We tabulated the correlation to make it more visual. Please refer to p.8.

Reviewer 2 Report

This study presented an analysis of the types and characteristics of rehabilitation specialists’ experience in administering virtual reality programs. Based on the Q methodology, the research suggested ways to improve VR programs and laid a foundation for practical directions in program administration. It is better to address the following issues.

1.       Figure 1 should be adjusted to avoid text coverage.

2.       The styles of the tables should be normalized and unified.

Author Response

1) Figure 1 should be adjusted to avoid text coverage.

Authors response: We revised accordingly. Please refer to p.4.

2) The styles of the tables should be normalized and unified.

Authors response: We normalized and incorporated the style of the table according to your comment. Please refer to p.5 to 11.

Reviewer 3 Report

The purpose of this study was to examine the nature and characteristics of rehabilitation specialists' virtual reality program administration experiences. A literature review and a thorough interview with rehabilitation experts were used to derive thirty-three statements as claimed by the authors. QUANL was used to analyze the data. According to the authors, this study is important because it makes recommendations for how to make virtual reality programs for people with disabilities better with the help of rehabilitation specialists.

I put my observations as under:

1.      The case for using Q methodology is tenuous.

2.      Figure 1 cannot be read.

3.      Please offer a figure that details each step of the Q-methodology.

4.      Is a sample of 22 reliable? Please check again. If not, kindly give a compelling explanation.

5.      It is important to explain why only a 9-point Likert scale was used.

6.      There are no literature reviews to be found here. Please compare this with at least 20 new works that you add. Please also include a comparison table.

7.      Please describe the limitations of the current study and the directions for future research.

8.      How were the 33 statements derived from the literature review? Please outline the entire set of requirements.

Author Response

1)      The case for using Q methodology is tenuous.

Authors’ response: Thank you for your comment. We have added more details in the methods section as advised.

2) Figure 1 cannot be read.

Authors’ response: We revised Figure 1 according to your comment. Please refer to p.4.

3)      Please offer a figure that details each step of the Q-methodology.

Authors’ response: We described each step in detail in Figure 1. Please refer to p.4.

4)      Is a sample of 22 reliable? Please check again. If not, kindly give a compelling explanation.

Authors’ response: We provided a convincing explanation in the text. Please refer to p.5 and p.6.

5)      It is important to explain why only a 9-point Likert scale was used.

Authors’ response: We have included an explanation for why we used the 9-point Likert scale. Please refer to p.7.

6)      There are no literature reviews to be found here. Please compare this with at least 20 new works that you add. Please also include a comparison table.

Authors’ response: We have added more literature reviews as advised. Please refer to p.11 to 12.

We have also created a comparison table for your reference as shown below.

Before

After

Type 1 rehabilitation specialists said that even though programs should be provided based on the consideration of various factors, including participants' functional level and disability type, current programs administered to the disabled were not specialized for them and so, it was not possible to provide programs customized for disabled individuals. Accordingly, to leverage the strengths of VR programs while decreasing risk and ultimately, to contribute to the development of rehabilitation and social return of people with disabilities via VR programs, the needs of disabled individuals should be identified via usability assessment (e.g., using the System Usability Scale or performing focus group interview (FGI)) and needs-based VR programs be developed.

second, type 2, labelled "emphasis on experts' role in paying attention continuously and their experience," was of opinion that rehabilitation specialists should be sensitive because disabled individuals' attitude toward program participation varies according to their condition and that VR programs should be administered by highly experienced therapists. In addition, type 2 thought that rehabilitation specialists' experience was important in VR program application and participation in VR programs required consistency and continuity, accordingly the specialists should continue to pay attention to their patients. In other words, rehabilitation specialists in type 2 seemed to think that the expertise of therapists was essential for administering VR programs to disabled individuals and their role was critical. This finding is consistent with previous study findings that the levels of rehabilitation specialists and service were directly linked with each other [50]. In addition, personal characteristics of people with disabilities using rehabilitation centers somewhat strongly influenced rehabilitation specialists' competency, as well as social and regulatory environments [51].

Type 2 rehabilitation specialists stated that because the therapists' experience would impact disabled individuals participating in VR programs, highly experienced and adequately educated therapists should administer the programs. Hence, to improve program quality, rehabilitation specialists should be educated to increase expertise in VR program administration. Furthermore, requirements to nationally register and certify practitioners in order to be qualified and administer programs to people with disabilities should be considered.

Type 1 rehabilitation specialists said that even though programs should be provided based on the consideration of various factors, including participants’ functional level and disability type, current programs administered to the disabled were not specialised for them and so, it was not possible to provide programs customised for disabled individuals. Accordingly, to leverage the strengths of VR programs while decreasing risk and ultimately, to contribute to the development of rehabilitation and social return of people with disabilities via VR programs, the needs of disabled individuals should be identified via usability assessment (e.g. using the System Usability Scale or performing focus group interview (FGI)) and needs-based VR programs be developed. This is consistent with previous studies showing high levels of satisfaction as a solution for rehabilitation and treatment of people with brain lesion and securing positive and effective data for patients through continuous clinical trials and usability evaluation [44]. In addition, it is consistent with previous studies that showed virtual reality technology in specialised rehabilitation areas for people with stroke and Parkinson's disease and helping their daily life [45-49].

Second, type 2, labelled ‘emphasis on experts’ role in paying attention continuously and their experience’, was of opinion that rehabilitation specialists should be sensitive because disabled individuals’ attitude toward program participation varies according to their condition and that VR programs should be administered by highly experienced therapists. In addition, type 2 thought that rehabilitation specialists’experience was important in VR program application and participation in VR programs required consistency and continuity, accordingly the specialists should continue to pay attention to their patients. In other words, rehabilitation specialists in type 2 seemed to think that the expertise of therapists was essential for administering VR programs to disabled individuals and their role was critical. This finding is consistent with previous study findings that the levels of rehabilitation specialists and service were directly linked with each other [50]. In addition, personal characteristics of people with disabilities using rehabilitation centers somewhat strongly influenced rehabilitation specialists’ competency, as well as social and regulatory environments [51]. Furthermore, it is consistent with previous studies in which the relationship between people with disabilities and the experts providing learning to the people with disabilities influenced motivation and motivation influenced learning [52].

Type 2 rehabilitation specialists stated that because the therapists’ experience would impact disabled individuals participating in VR programs, highly experienced and adequately educated therapists should administer the programs. Hence, to improve program quality, rehabilitation specialists should be educated to increase expertise in VR program administration. Furthermore, requirements to nationally register and certify practitioners in order to be qualified and administer programs to people with disabilities should be considered. This is consistent with the findings that the role of competent supervisors and experts plays a decisive role in the rehabilitation of the people with disabilities [53], and shows that services are more effective when provided through the cooperation of experts in each field for them [54].

7)      Please describe the limitations of the current study and the directions for future research.

Authors’ response: We gave information on the limitations of the study. Please refer to p.13.

8)      How were the 33 statements derived from the literature review? Please outline the entire set of requirements.

Authors’ response: We explained how the statement was derived. Please refer to p.4.

Reviewer 4 Report

The paper is well written and provides insights from rehabilitation specialists into ways of improving VR programs for rehabilitation. The authors describe a sound methodology and analysis used to form the conclusions of their study. The limitations of the study are also provided.

There needs to be a section on relate work to describe what research others have conducted in the area and to explain how this work is different from related work.

The paper mentions that the 33 statements were derived based on a literature review. It'll be nice to have some way of knowing what research literature was referred to in the study. 

As the authors point out in the Limitations section, the demographics of the 22 specialists were not sufficiently discussed. The paper needs to at least provide a brief overview of the participants. Were the specialists from a particular organization? gender? do they work with adults, with children, or the elderly?  

Figure 1 - text in the boxes are covered by other boxes.

Author Response

1) There needs to be a section on relate work to describe what research others have conducted in the area and to explain how this work is different from related work.

Authors response: We revised the introduction according to your comment. Please refer to p.1 and p.2.

2) The paper mentions that the 33 statements were derived based on a literature review. It'll be nice to have some way of knowing what research literature was referred to in the study. 

Authors response: We wrote the reference according to your comment. Please refer to p.4.

3) As the authors point out in the Limitations section, the demographics of the 22 specialists were not sufficiently discussed. The paper needs to at least provide a brief overview of the participants. Were the specialists from a particular organization? gender? do they work with adults, with children, or the elderly? 

Authors response: We have outlined a brief summary of 22 experts according to your comments. Please refer to p.5 and p.6.

4) Figure 1 - text in the boxes are covered by other boxes.

Authors response: We revised Figure 1 according to your comment. Please refer to p.4.

Round 2

Reviewer 3 Report

Thanks. I can see that this manuscript has undergone significant revision. I'm content.